# Piloting competency assessments for an evidence-based brief psychological intervention with Arabic-speaking non-specialists in Switzerland

## Research Article

EQUIP; competency assessment; mental health psychosocial support (MHPSS); Problem Management Plus (PM+); non-specialist mental health care

**Corresponding author:**
Mahmoud Hemmo;
Email: mahmoud.hemmo@usz.ch

Mahmoud Hemmo[1,2] ⬤, Aemal Akhtar[3,4] ⬤, Brandon A. Kohrt[5] ⬤,
Gloria Pedersen[5,6,7] ⬤, Abdul Fattah Alkamel[1], Chantal Martin Sölch[2] ⬤,
Alison Schafer[8] ⬤, Julia Spaaij[1] ⬤, Richard Bryant[3] ⬤ and Naser Morina[1] ⬤

[1]Department of Consultation-Liaison Psychiatry and Psychosomatic Medicine, University Hospital of Zurich, University of Zurich, Zurich, Switzerland; [2]Department of Psychology, University of Fribourg, Fribourg, Switzerland; [3]School of Psychology, UNSW Sydney, Sydney, Australia; [4]Division of Insurance Medicine, Department of Clinical Neuroscience, Karolinska Institute, Stockholm, Sweden; [5]Center for Global Mental Health Equity, The George Washington University, Washington, DC, USA; [6]Department of Global Health and Social Medicine, Harvard Medical School, Boston, MA, USA; [7]Mental Health Program, Partners in Health, Boston, MA, USA and [8]Department of Mental Health, Brain Health and Substance Use, World Health Organization, Geneva, Switzerland

## Abstract

The global challenge of closing the treatment gap highlights the need for innovative interventions. Problem Management Plus (PM+), developed by the World Health Organization (WHO), is an evidence-based brief psychological intervention designed to address this gap by involving non-specialist helpers. In this study, 'non-specialists' or 'helpers' are individuals without formal training in mental health, who have been trained in and have been delivering individual PM+ for more than 1.5 years. To enhance quality in mental health care, especially with non-specialists, WHO and the United Nations International Children's Emergency Fund (UNICEF) have launched the Ensuring Quality in Psychosocial and Mental Health Care (EQUIP) platform, an open-access resource for competency-based training. This study evaluates the acceptability and preliminary utility of EQUIP assessment tools. Thirteen helpers were assessed using the ENhancing Assessment of Common Therapeutic Factors (ENACT) and the PM+ assessment tool, culturally adapted and translated for Arabic-speaking helpers in Switzerland. The results indicate that the EQUIP tools can identify strengths and areas for improvement, provide valuable feedback for training, and thus have great potential for enhancing mental health care quality.

الملخص

إن الفجوة العالمية في الحصول على العلاج النفسي تبرز الحاجة المُلِحّة إلى تطوير تدخلات مبتكرة. ويُعدّ برنامج الإدارة المطورة للمشكلات (Problem Management Plus (PM+) ، الذي طورته منظمة الصحة العالمية، أحد التدخلات النفسية الموجزة المبنية على الأدلة التي تهدف إلى معالجة هذه الفجوة، وذلك من خلال إشراك مساعدين غير متخصصين في تقديم الرعاية. يُشير مصطلح "غير المتخصصين" في هذه الدراسة إلى الأفراد الذين لا يمتلكون تدريبًا رسميًا في مجال الصحة النفسية، والذين خضعوا لتدريب على تقديم جلسات PM+ الفردية، ويقومون بممارستها منذ أكثر من عام ونصف.

وفي سبيل تحسين جودة الرعاية النفسية، ولا سيما عند الاعتماد على غير المتخصصين، أطلقت منظمة الصحة العالمية بالتعاون مع منظمة الأمم المتحدة للطفولة (اليونيسف) منصة ضمان الجودة في الدعم النفسي الاجتماعي والرعاية الصحية النفسية (Ensuring Quality in Psychosocial and Mental Health Care (EQUIP))، وهي منصة مفتوحة المصدر تُقدم موارد تدريبية قائمة على الكفاءة.

تهدف هذه الدراسة إلى تقييم مدى قبول وجدوى أدوات التقييم المقدمة عبر منصة EQUIP. حيث قد تم تقييم ثلاثة عشر مساعدًا باستخدام أداتين: "تعزيز تقييم العوامل العلاجية المشتركة" (ENhancing Assessment of Common Therapeutic Factors (ENACT) وأداة تقييم الأداء الخاصة بـ PM+، وذلك بعد إجراء تكييف ثقافي وترجمة للأداتين لتناسب المساعدين الناطقين بالعربية في سويسرا.

أظهرت النتائج أن أدوات EQUIP تُمكّن من تحديد مكامن القوة ومجالات التحسين، كما تتيح تقديم تغذية راجعة بنّاءة لأغراض التدريب. وتشير هذه النتائج إلى الإمكانات الكبيرة التي تنطوي عليها هذه الأدوات في تحسين جودة خدمات الرعاية النفسية.





## Impact statement

World Health Organization (WHO) and United Nations Children's Fund (UNICEF) have launched an open-access web platform called EQUIP. Using a competency-based approach, this platform provides standardized tools and materials in multiple languages to improve training and

monitoring of mental health care providers worldwide. EQUIP tools are continuously being supplemented, translated into further languages, and culturally adapted. The aim of this pilot study was to assess acceptability and utility of the Arabic EQUIP tools for use with individual PM+ (Problem Management Plus) helpers in Switzerland. Results showed that the tools were well received and provided useful and valuable feedback. Use of these tools in real-world settings shows potential to improve mental health care in other settings, which may include resource-constrained contexts or where services are lacking for groups with particular vulnerabilities. Further research in larger samples and different contexts is warranted to better understand this potential. These findings are relevant to all stakeholders involved in psychosocial interventions in Arabic, particularly, and the EQUIP tools show promise in improving quality of care and treatment outcomes for clients. An expansion of the application and further studies in Switzerland and the Middle East are underway.

## Introduction

Integrating evidence-based brief psychological interventions into primary and secondary healthcare is a promising step toward narrowing the treatment gap (Kohn et al., 2004; Saxena et al., 2007; Patel et al., 2016; Moitra et al., 2022). Problem Management Plus (PM+), developed by the World Health Organization (WHO), is a prominent example of evidence-based brief psychological intervention delivered by non-specialists, also known as helpers (Dawson et al., 2015). In this study, 'non-specialists' or 'helpers' refer to individuals who are not formally trained mental health professionals, but who have been trained and have gained experience in delivering PM+.

Akin to other evidence-based brief psychological interventions, PM+ combines advantages of scalability, adaptability, and well-structured delivery by trained non-specialists. However, it could be argued that PM+ has stronger potential for implementation in resource-constrained settings. PM+ is less limited to specific conditions (*e.g.*, Group Interpersonal Therapy (IPT) for depression (World Health Organization (WHO) and Columbia University, 2016) or groups (*e.g.*, Thinking Healthy for prenatal women (World Health Organization (WHO), 2015). PM+ may also face fewer implementation barriers in low-resource settings, given it is an open-science resource, which increases its accessibility (Watts et al., 2020; Cuijpers et al., 2024).

PM+ is designed for individuals with common mental health problems (*e.g.*, anxiety, stress, depressive symptoms) following adversity. This transdiagnostic intervention has been shown to be effective in numerous randomized controlled trials (RCTs) in low- and middle-income countries (LMICs) (Rahman et al., 2016; Bryant et al., 2017, 2022; Jordans et al., 2021) and in high-income countries (HICs) (De Graaff AM et al., 2023; Spaaij et al., 2023a). These findings were supported by a scoping review reporting moderate-to-large effect sizes for PM+ in reducing symptoms of depression, anxiety, psychological distress, functional impairment, and PTSD (Mwangala et al., 2024). Additionally, a meta-analysis found small to moderate positive effects across these and other mental health symptoms, as well as a small effect for improvements in well-being/quality of life and social support/relationships compared with (enhanced) care-as-usual (Schäfer et al., 2023).

PM+ consists of five sessions, in which clients learn four psychological strategies (Managing Stress, Managing Problems, Behavioral Activation, Strengthening Social Support), receive psychoeducation and information on relapse prevention (Dawson et al., 2015) and can be delivered individually or in groups (World Health Organization (WHO), 2016b, 2020). This study refers to individual PM+.

To ensure quality in non-specialist care, the WHO recommends competency-based approaches to training and supervision (World Health Organization (WHO), 2022). Competency refers to "the observable ability of a person, integrating knowledge, skills, and attitudes in their performance of tasks" and "competencies are durable, trainable and through the expression of behaviors, measurable" (Mills et al., 2020). In competency-based training, learners are evaluated to track the development of skills and knowledge, with iterative modifications to training to assure that the learners achieve the minimum level of competency needed to perform their clinical tasks (Frank et al., 2010). Research on professionals in HICs shows that more competencies lead to better treatment (Imel and Wampold, 2008; Sparks et al., 2008; Karson and Fox, 2010; Wampold, 2015). This is also likely to be the case for helpers. However, until recently, there has been insufficient data on standardized competency monitoring in helpers (Singla et al., 2017, 2021).

To address the lack of competency assessments and monitoring among helpers, WHO and United Nations International Children's Emergency Fund (UNICEF) jointly launched the Ensuring Quality in Psychosocial and Mental Health Care (EQUIP) platform (Kohrt et al., 2020, 2025), in March 2022. EQUIP is an open-access web resource offering training resources, implementation aids, and assessment tools in multiple languages to develop and monitor competencies of helpers and support a competency-based approach to training and supervision to enhance quality of care delivery (www.equipcompetency.org). Key components of the EQUIP competency-based approach include selection of defined competencies, scripted role-plays, standardized assessments, and feedback. EQUIP offers competency assessment tools for common therapeutic factors in psychological services, including the ENACT (ENhancing Assessment of Common Therapeutic factors) assessment tool (Kohrt et al., 2015a), accessible in 14 languages (Kohrt et al., 2025) and treatment-specific competency tools that can be tailored to brief psychological interventions, including PM+. EQUIP is designed for flexible use by trainers, supervisors, and other personnel involved in mental health and psychosocial support (MHPSS) interventions across different cultures and contexts (Kohrt et al., 2020, 2025).

EQUIP assessment tools have been examined in various settings (Kohrt et al., 2025), including the ENACT tool (Kohrt et al., 2015a, 2015b; Gebrekristos et al., 2021; Pedersen et al., 2021, 2023b; Jordans et al., 2022; Mwenge et al., 2022) and the PM+ competency tool (Gebrekristos et al., 2021; Pedersen et al., 2021). These studies show that the tools possess sufficient inter-rater reliability, are sensitive to change, provide reliable results, and support tailoring of training measures. Nonetheless, further exploration of their psychometric properties and cost-effectiveness across various contexts is needed, including its implementation in different languages (Kohrt et al., 2015a; Jordans and Kohrt, 2020; Pedersen et al., 2020, 2021).

The practical competencies required of helpers delivering PM+ suggest that competency-based assessments are essential to identify training needs and ensure high-quality care. Therefore, the EQUIP platform also provides 12 PM+ competency assessment items for flexible use (Supplementary S1). The forthcoming WHO PM+ training manual is planned to include a selection of 9 PM+ competency items (Supplementary S2). EQUIP has been tested, using

the ENACT assessment tool alongside a PM+ tool during a training of helpers in Ethiopia (Gebrekristos et al., 2021; Pedersen et al., 2021). It was also evaluated in a remote training of helpers in Africa, Europe, and USA (McBride et al., 2021), a training of trainers and training of helpers in Nepal (Pedersen et al., 2023a), and in a PM + A (adapted for clients with alcohol misuse) RCT in Uganda (Van Der Boor et al., 2024).

A cost analysis of an EQUIP-based approach to delivering PM+ training of trainers and helpers shows potential benefits for improving helper competencies and client outcomes, while time and cost differences as compared to a standard PM+ training are minimal (Pedersen et al., 2023a), reinforcing the importance of evaluating EQUIP in high-income settings. This paper presents findings from the application of EQUIP assessment tools within the context of a larger RCT, the Scaling up psychological interventions with Syrian Refugees (STRENGTHS) project (Sijbrandij et al., 2017). One of STRENGTHS' objectives was to address mental health needs of refugees and asylum seekers (RAS) in Switzerland, specifically through the implementation of PM+ (Spaaij et al., 2023a). Approximately, 22,000 Syrian RAS currently live in Switzerland (Staatssekretariat für Migration (SEM), 2024). RAS in Switzerland face several structural and socio-cultural barriers to accessing mental healthcare, which has also been shown true for the Syrian population (Kiselev et al., 2020). In HICs, RAS also experience discriminatory behavior on the part of mental health professionals, contributing to existing obstacles to appropriate care (Dumke et al., 2024). Therefore, delivery of brief psychological interventions delivered by helpers has potential to address an important public mental health gap.

This pilot study examines acceptability and preliminary utility of the adapted EQUIP competency assessment tools in evaluating competencies of Arabic-speaking helpers delivering PM+ in Switzerland. This is one of the first studies evaluating EQUIP in a HIC. The study also aims to assess the tools' ability to identify helpers' inter- and intra-individual strengths and areas for improvement. Additionally, we explore their perceived usefulness in informing subsequent competency-based training. We hypothesize that the EQUIP assessments will be considered acceptable, useful, and helpful in improving training activities.

## Methods

### Setting

This study was conducted at the University Hospital Zurich (USZ) in January 2020 as part of the STRENGTHS trial in Switzerland and was approved by the Ethics Committee of the Canton of Zurich (BASEC Nr. 2017-0117).

### Sample

Thirteen PM+ helpers were evaluated using EQUIP. Helpers' participation in the competency assessment was mandatory as part of their job responsibilities. Selection criteria for helpers to participate in STRENGTHS included migration to Switzerland after 2015 and proficiency in Arabic and either German or English (Spaaij et al., 2022). In line with recruitment strategies in Switzerland, additionally a higher education diploma was required to uphold high training and implementation standards. Similarly, a large-scale implementation project, integrating PM+ nationwide into the health care system, prioritizes well-educated candidates (Spaaij et al., 2023b).

Prior to the assessment, all helpers had received 8 days of in-class training led by two experienced Arabic-speaking PM+ trainers and completed introductions to Good Clinical Practice and Psychological First Aid. They then underwent an exercise phase, delivering PM+ to two practice clients under close supervision, before working with real clients, who were Syrian RAS with common mental health problems. By the time of this study's assessment, each helper had approximately 1.5 years of experience delivering PM+. No formal pre-assessment was conducted before this study. Structured training and regular supervision allowed continuous monitoring of their competencies.

### Measures

The first tool was ENhancing Assessment of Common Therapeutic factors (ENACT) (Kohrt et al., 2015a), which assesses foundational helping skills in working with adults. The second tool was the PM+ competency assessment tool (Pedersen et al., 2021). Items for both tools (Table 1) are qualitatively assessed across four levels. Level 1 is a critical red flag for unhelpful behavior that is potentially harmful to people receiving PM+ support. Level 2 indicates a need for improvement as not all basic skills are mastered. Level 3 reflects competence in essential skills necessary for effective delivery. Finally, Level 4 indicates mastery of a competency, indicating that the helper not only meets all basic requirements (*i.e.*, Level 3), but also excels by incorporating advanced skills.

The ENACT tool, developed with 18 items for use in psychological interventions, including settings that involve helpers, was originally piloted with primary care workers trained in the Mental Health Gap Action Programme (World Health Organization (WHO), 2016a) and in psychological interventions, in Nepal (Kohrt et al., 2015a). For EQUIP, ENACT was adapted into 15 items (Pedersen et al., 2020), each including three sets of attributes, where each attribute is accompanied by a checkbox (Figure 1). If no or not all attributes in Levels 2 or 3 (*e.g.*, Appropriately encourages client to share feelings) are demonstrated, the item is scored as Level 2. If all Level 2 and 3 attributes are displayed, the item is scored at Level 3. If all Level 2 and 3 attributes and at least one Level 4 attribute (*e.g.*, Asks client to reflect on the experience of sharing emotions) are observed during the assessment role-plays, the assessment is rated as a Level 4. If at least one attribute in Level 1 (*e.g.*, Forces client to describe emotions) is evident during the assessment (*i.e.*, any unhelpful behavior, irrespective of whether attributes were demonstrated in Levels 2, 3 or 4), the item is assessed as Level 1.

The PM+ tool was originally created for implementation in Ethiopia and this study, wherein both implementing sites had eight items in common, covering competencies supporting components of the PM+ intervention: Managing Problems, Managing Stress, and Get Going, Keep Doing (Pedersen et al., 2021). That version of the tool did not capture all 5 PM+ sessions and their core strategies. To close this gap, two items were added based on expert consensus: Strengthening Social Support and Staying Well and Looking to the Future. These additions ensure evaluation of all core competencies essential for effective PM+ implementation. The 10-item tool used in this study finally included five items on introducing Managing Problems, four 'Review' items on the PM+ core strategies (*i.e.*, Managing Stress, Managing Problems, Get Going, Keep Doing and Strengthening Social Support) and one item on Staying Well and Looking to the Future. Review items were created to allow for compact assessment in reduced time. The PM+ tool for this study differs slightly from EQUIP's most recent PM+ tool version and the

**Table 1.** EQUIP tool item descriptions

| Item No. | Title | Description |
|---|---|---|
| ENACT tool items 1–15 | | |
| 1 | Collaborative Goal Setting & Addressing Client's Expectations | Helper asks about the client's goals and discusses expectations for treatment. |
| 2 | Promotion of Realistic Hope for Change | Helper builds the client's hope for change, helping them feel positive about the future and what can be achieved in treatment. |
| 3 | Explanation and Promotion of Confidentiality | Helper ensures confidentiality and explains potential breaches (*e.g.*, harm to self). |
| 4 | Assessment of Harm to Self, Harm to Others, Harm from Others & Developing Collaborative Response Plan | The helper assesses for harm to self, harm to others, or harm from others. If needed, they create a safety plan together. |
| 5 | Appropriate Involvement of Family Members and Relevant Others | Helper considers the client's intent to include a family member or other close person in the treatment. |
| 6 | Connection to Social Functioning & Impact on Life | Helper examines how mental health concerns affect the client's daily life. |
| 7 | Incorporation of Coping Mechanisms & Prior Solutions | The helper identifies and develops the client's existing coping skills and ways of problem-solving abilities. |
| 8 | Exploration of Client's & Social Support Network's Explanation for Problem (Causal & Explanatory Models) | Helper explores the client's and their social environment's understanding of the causes and consequences of their problems. |
| 9 | Non-Verbal Communication & Active Listening | Helper uses culturally appropriate non-verbal cues and active listening to show engagement with the client. |
| 10 | Rapport Building and Self-Disclosure | Helper builds rapport and strengthens the relationship with the client, starting with a warm welcome and introduction. |
| 11 | Demonstration of Empathy, Warmth & Genuineness | Helper expresses a sincere understanding of the client's feelings and thoughts, providing warmth, free from personal judgments. |
| 12 | Verbal Communication Skills | By using open-ended questions, summarizing/clarifying, the helper communicates effectively and shows understanding and support. |
| 13 | Exploration & Normalization of Feelings | The helper explores the client's feelings, normalizes their experience and reminds them that it's shared by others. |
| 14 | Elicitation of Feedback When Providing Advice, Suggestions & Recommendations | When giving advice or making suggestions, the helper asks for feedback from the client. |
| 15 | Psychoeducation and Use of Local Terminology | Helper uses local language and psychological concepts to provide psychoeducation. |

*(Continued)*

**Table 1.** (*Continued*)

| Item No. | Title | Description |
|---|---|---|
| PM+ tool items 1–10 | | |
| 1 | Managing Problems: Listing and Choosing Problems | Helper and client list manageable problems together. The helper guides the client in selecting one to start with, ideally an easy problem. |
| 2 | Managing Problems: Defining the Problem | Helper and client define the selected problem together. The helper ensures that the problem is specific, practical and contains elements that can be influenced or controlled. |
| 3 | Managing Problems: Brainstorming Solutions | The helper encourages the client to come up with as many ideas as possible to solve or change (part of) the problem. |
| 4 | Managing Problems: Choosing A Solution | Helper guides the client in evaluating the brainstormed solutions and in choosing one achievable, helpful solution, considering both consequences. |
| 5 | Managing Problems: Solutions Action Plan | The helper guides the client in creating a step-by-step action plan to solve the problem, setting specific times and using reminders. |
| 6 | Review of Stress Management and Relaxation | The helper reviews the client's practice of the breathing exercise, discusses challenges and helps to find ways to overcome them, encouraging continued regular use. |
| 7 | Review of Managing Problems | The helper reviews the client's implementation of the action plan for the selected problem, discusses outcomes and challenges. If needed, the helper guides in adapting it and encourages continued regular use. |
| 8 | Review of Get Going, Keep Doing | The helper reviews the client's implementation of planned activities, discusses achievements and challenges, and encourages the selection of new activities or tasks for continued practice. |
| 9 | Review of Strengthening Social Support | The helper reviews the client's implementation of the planned social support activities, discusses achievements and challenges, and encourages the identification of additional ways to strengthen their social support. |
| 10 | Staying Well & Looking to the Future | The helper explores the client's future goals and helps them to imagine how they can help others using the strategies they have learned. Helper explains concepts of recovery and relapse prevention, encouraging continued use of these strategies to maintain wellbeing. |

ENACT tool in their rating procedure. Here, PM+ items comprise four levels of attribute descriptions without checkboxes (Figure 2). The rater selects the level matching the observed performance most accurately based on overall impression. Level 1 is scored if at least one Level 1 attribute is evident.

| Unhelpful or potentially harmful behaviors | Basic helping skills | Advanced helping skills |
|---|---|---|
| ❑ Makes statements that client's response is unusual or atypical for others in similar situations (e.g., people don't usually react this way) <br> ❑ Minimizes or dismisses client's feelings or emotions <br> ❑ Forces client to describe emotions | ❑ Appropriately encourages client to share feelings <br> ❑ Explain that others may share similar symptoms, reactions, and concerns, given similar experiences | ❑ Explores potential reasons for reluctance to share emotions <br> ❑ Asks client to reflect on the experience of sharing emotions <br> ❑ Validates emotional responses while also reframing potential harmful emotional reactions |

Check the level that best applies (only one level should be checked)

☐ **Level 1** *any unhelpful behavior*  ☐ **Level 2** *no basic skills, or some but not all basic skills*  ☐ **Level 3** *all basic skills*  ☐ **Level 4** *all basic helping skills plus any advanced skill*

**Figure 1.** ENACT competency assessment item – Exploration & Normalization of Feelings.

| Level 1 | Level 2 | Level 3 | Level 4 |
|---|---|---|---|
| Scolds or blames beneficiary for potential 'downs' in future (e.g. 'If you will feel distress again, it means you were lazy'); Lectures client; Doesn't react with empathy to beneficiary's fears about the future | Does not explain the concept of recovery and relapse prevention at all; Does not suggest beneficiary to use strategies in future | Explains the concept of recovery and relapse prevention; Reviews potential future 'downs' and adequate use of strategies; Defines concrete future goal(s) and an action plan with beneficiary | ACHIEVES LEVEL 3 and uses metaphoric language; Reflects on process of learning strategies; Discusses one case example to help imagining how to help others; Encourages and enforces beneficiary's confidence to use strategies in future; Plans use of reminders |

**Figure 2.** PM+ competency assessment item – Staying Well and Looking to the Future.

To ensure cultural and contextual relevance in our study, ENACT and the adapted PM+ tool were translated into Arabic following the model by Van Ommeren et al. (1999), comprising steps to maintain comprehensibility, acceptability, relevance and completeness, including translation and blinded back-translation by two bilingual health experts, and evaluation of both the original and back-translation by a third bilingual expert. Tool paper versions in DIN A3 size were used for the rater to have a better view of all items.

EQUIP assessment tools involve a rater evaluating role-play interactions between the helper and a trained actor portraying a service user (Pedersen et al., 2021). The actor in our study, a research team member and health professional with Syrian background, played a key role in adapting and translating the EQUIP tools to ensure their authenticity. He completed the EQUIP e-learning modules for actors (www.equipcompetency.org) which included in-depth discussions of assessment items and achieving an understanding of the rating process. He also had previously been involved in translating further PM+ materials, which served his familiarity with the topic.

A single actor-case-vignette was developed based on clinical experience with Syrian RAS in mental healthcare and individual PM+ in Switzerland, detailing the client's life situation, stressors, promptings, and suggested actor responses (Supplementary S3). To prevent bias from discussions among helpers while maintaining consistency, structured variations (e.g., stressors, strategy implementation) were applied in real-time.

This study involved one senior EQUIP rater (Author), an experienced psychologist and PM+ expert. He was involved in the development of EQUIP and played a key role in tool adaptation for this study. As a trainer and supervisor, he provided both the initial training and the ongoing supervision of the PM+ helpers in this study.

We have applied a standardized approach for providing verbal, competency-based feedback to the helpers, including summarizing overall performance, highlighting two to three areas for improvement, and stating the most noteworthy competencies successfully demonstrated.

In EQUIP assessments, role plays can be evaluated either immediately through direct observation or at a later stage using video or audio recordings. In this study, we opted for real-time assessment to provide helpers with immediate feedback. Still, for potential future re-evaluation, assessments were videotaped.

## Assessment protocol

The assessment, including feedback, required 65 minutes per helper (Supplementary S4). Initially, the rater explained to helpers aims and procedure of the assessment, followed by informed consent.

The rater instructed the first ENACT role-play ("You are a helper, meeting the client for the first time. You have 15 minutes to learn about the individual's distress."), after which the helper initiated a mock session with the actor. During ENACT, the actor introduced Goal Setting, Promotion of Realistic Hope, Suicidal Ideation, Confidentiality and Involving Family, prompting them (*e.g.* "There are moments when I think about killing myself"), up to three times each, if necessary. After concluding the ENACT role-play, the rater instructed the helper to conduct a 10-minutes multi-item PM+ role-play on Managing Problems, followed by five single-item PM+ role-plays, 5 minutes each (Supplementary S4). Following the assessment, the helper left the room, and feedback was discussed and finalized between the rater and actor to enhance accuracy of the assessment and reduce bias. Once completed, the actor left the room, and the helper was invited back to share their reflections and receive feedback from the rater.

## Analyses

We assigned each item a nominal value based on observed performance according to its assessed level:

1. Unhelpful or potentially harmful behaviors;
2. No basic skills OR Some but not all basic skills;
3. All basic skills;
4. Advanced helping skills.

Descriptive statistics were generated to provide an overview of helpers' competencies in the ENACT (Table 2) and PM+ (Table 3) assessments, presenting individual items and summed scores across all items. For streamlined reporting, Levels 3 and 4 were combined to the category 'competent level'. We compared items scored at competent level to those marked as "no or not all basic skills" (Level 2) or "unhelpful" (Level 1). The data were analyzed using IBM SPSS Statistics (version 29).

Furthermore, we collected qualitative feedback from both the rater and actor on EQUIP's perceived utility, including observations the rating process, acting task, delivery of competency-based feedback, and their impression of helpers' acceptance of the assessment protocol based on their reactions.

## Results

### Participants

Thirteen helpers, including seven women (53.85%) and six men (Mean age = 37.7 years, SD = 7.1), were evaluated with EQUIP following 1.5 years of PM+ delivery. Ten held bachelor's degrees,

**Table 2.** Aggregated competency scores for the ENACT assessment

| Item | ENACT item | Level 1 | Level 2 | Level 3 | Level 4 | Level 3 + 4 | Mean (SD) | Median | Range |
|------|-----------|---------|---------|---------|---------|-------------|-----------|--------|-------|
| 1 | Collaborative Goal Setting & Addressing Client's Expectations | 0 | 2 | 5 | 6 | 11 | 3.31 (0.75) | 3.00 | 2.00 |
| 2 | Promotion of Realistic Hope for Change | 0 | 3 | 5 | 5 | 10 | 3.15 (0.80) | 3.00 | 2.00 |
| 3 | Explanation and Promotion of Confidentiality | 0 | 0 | 0 | 13 | 13 | 4.00 (0.00) | 4.00 | 0.00 |
| 4 | Assessment of Harm to Self, Harm to Others, Harm from Others & Developing Collaborative Response Plan | 1 | 5 | 6 | 1 | 7 | 2.54 (0.78) | 3.00 | 3.00 |
| 5 | Appropriate Involvement of Family Members and Relevant Others | 1 | 6 | 2 | 4 | 6 | 2.69 (1.03) | 2.00 | 3.00 |
| 6 | Connection to Social Functioning & Impact on Life | 0 | 4 | 9 | 0 | 9 | 2.69 (0.48) | 3.00 | 1.00 |
| 7 | Incorporation of Coping Mechanisms & Prior Solutions | 1 | 8 | 1 | 3 | 4 | 2.46 (0.97) | 2.00 | 3.00 |
| 8 | Exploration of Client's & Social Support Network's Explanation for Problem (Causal & Explanatory Models) | 0 | 13 | 0 | 0 | 0 | 2.00 (0.00) | 2.00 | 0.00 |
| 9 | Non-Verbal Communication & Active Listening | 0 | 1 | 1 | 11 | 12 | 3.77 (0.60) | 4.00 | 2.00 |
| 10 | Rapport Building and Self-Disclosure | 0 | 7 | 4 | 2 | 6 | 2.62 (0.77) | 2.00 | 2.00 |
| 11 | Demonstration of Empathy, Warmth & Genuineness | 2 | 2 | 9 | 0 | 9 | 2.54 (0.78) | 3.00 | 2.00 |
| 12 | Verbal Communication Skills | 1 | 1 | 1 | 10 | 11 | 3.54 (0.97) | 4.00 | 3.00 |
| 13 | Exploration & Normalization of Feelings | 0 | 1 | 4 | 8 | 12 | 3.54 (0.66) | 4.00 | 2.00 |
| 14 | Elicitation of Feedback When Providing Advice, Suggestions & Recommendations | 0 | 10 | 2 | 1 | 3 | 2.31 (0.63) | 2.00 | 2.00 |
| 15 | Psychoeducation and Use of Local Terminology | 0 | 0 | 7 | 6 | 13 | 3.46 (0.52) | 3.00 | 1.00 |
| | **Total count of helpers per level score across all items** | **6** | **63** | **56** | **70** | **126** | **2.97 (0.20)** | **3.00** | **3.00** |
| | **Percentage of total count of helpers per level score across all items** | **3.08%** | **32.31%** | **28.72%** | **35.90%** | **64.62%** | | | |

This table presents the aggregated competency assessment scores applied in this study for the ENACT competency assessment among the helpers (N = 13). Values in the columns titled 'Level 1' through 'Level 3 + 4' indicate the number of helpers at each competency level for each item: Level 1 (Unhelpful or potentially harmful behaviors), Level 2 (No basic skills OR some but not all basic helping skills), Level 3 (All basic helping skills) and Level 4 (Advanced helping skills). Items rated at Level 3 or 4 were combined to represent a score for 'competent level'. The last three columns provide the corresponding statistical summaries (Mean, SD, Median, and Range). The last two rows present the total count of helpers per level score across all items.

**Table 3.** Aggregated competency scores for the PM+ assessment

| Item | PM+ item | Level 1 | Level 2 | Level 3 | Level 4 | Level 3 + 4 | Mean (SD) | Median | Range |
|------|----------|---------|---------|---------|---------|-------------|-----------|--------|-------|
| 1 | Managing Problems: Listing and Choosing Problems | 0 | 0 | 3 | 10 | 13 | 3.77 (0.44) | 4.00 | 1.00 |
| 2 | Managing Problems: Defining the Problem | 0 | 1 | 6 | 6 | 12 | 3.38 (0.65) | 3.00 | 2.00 |
| 3 | Managing Problems: Brainstorming Solutions | 0 | 2 | 6 | 5 | 11 | 3.23 (0.73) | 3.00 | 2.00 |
| 4 | Managing Problems: Choosing A Solution | 0 | 2 | 5 | 6 | 11 | 3.31 (0.75) | 3.00 | 2.00 |
| 5 | Managing Problems: Solutions Action Plan | 0 | 0 | 7 | 6 | 13 | 3.46 (0.52) | 3.00 | 1.00 |
| 6 | Review of Stress Management and Relaxation | 0 | 0 | 4 | 9 | 13 | 3.69 (0.48) | 4.00 | 1.00 |
| 7 | Review of Managing Problems | 2 | 0 | 2 | 9 | 11 | 3.38 (1.12) | 4.00 | 3.00 |
| 8 | Review of Get Going, Keep Doing | 0 | 0 | 1 | 12 | 13 | 3.92 (0.28) | 4.00 | 1.00 |
| 9 | Review of Strengthening Social Support | 0 | 0 | 2 | 11 | 13 | 3.85 (0.38) | 4.00 | 1.00 |
| 10 | Staying Well & Looking to the Future | 0 | 0 | 4 | 9 | 13 | 3.69 (0.48) | 4.00 | 1.00 |
| | **Total count of helpers per level score across all items** | **2** | **5** | **40** | **83** | **123** | **3.57 (0.29)** | **4.00** | **3.00** |
| | **Percentage of total count of helpers per level score across all items** | **1.54%** | **3.85%** | **30.77%** | **63.85%** | **94.62%** | | | |

This table presents the aggregated competency assessment scores applied in this study for the PM+ competency assessment among the helpers (N = 13). Values in the columns titled 'Level 1' through 'Level 3 + 4' indicate the number of helpers at each competency level for each item: Level 1 (Unhelpful or potentially harmful behaviors), Level 2 (No basic skills OR some but not all basic helping skills), Level 3 (All basic helping skills), and Level 4 (Advanced helping skills). Items rated at Level 3 or 4 were combined to represent a score for 'competent level'. The last three columns provide the corresponding statistical summaries (Mean, SD, Median, and Range). The last two rows present the total count of helpers per level score across all items.

two had master's degrees, and one had a certificate of higher education. Twelve helpers were Syrian refugees (92.31%), one was from Lebanon.

### ENACT performances

In the ENACT assessment (Table 1), helpers scored 64.62% of items as competent (Levels 3 and 4) and 3.08% as unhelpful (Level 1), (M = 2.97, SD = 0.20). All helpers demonstrated competency in Confidentiality and Psychoeducation. Additionally, at least half of the sample achieved competency (Median = 3) in Goal Setting, Promotion of Realistic Hope, Assessment of Harm, Impact on Life, Non-Verbal Communication, Empathy, Verbal Communication, and Exploration & Normalization of Feelings.

No helper achieved competency (Level 2) in Client's Explanation for Problem (*e.g.*, Asks about client's view on cause of problem). Unhelpful behaviors were observed twice in Empathy (*e.g.*, Critical of client's concerns) and once in Assessment of Harm (*e.g.*, Lectures client with religious or legal reasons against self-harm), Involvement of Family (*e.g.*, Tells client not to involve family or close person in any way during treatment or recovery), Coping & Prior Solutions (*e.g.*, Makes negative statements about client's coping mechanisms) and Verbal Communication (*e.g.*, Interrupts client).

### PM+ performances

94.62% of PM+ items (Table 3) were scored as competent (Levels 3 and 4), only 1.54% as unhelpful (Level 1) (M = 3.57, SD = 0.29). All helpers demonstrated competency in Listing/Choosing Problems, Solutions Action Plan, Review of Stress Management, Review of Get Going, Keep Doing, Review of Strengthening Social Support, and Staying Well & Looking Forward. At least half of the helpers (Median = 3) showed competency in all PM+ items.

One helper did not reach competency (Level 2) in Defining the Problem, while two lacked competency in Brainstorming and Choosing a Solution. Review of Managing Problems was rated unhelpful (Level 1) in two helpers.

### Qualitative feedback by rater and actor on the assessment

Rater and actor reported that they found the EQUIP package supportive in preparing for their roles and easy to understand. Timing and break planning were adequate and instructions for giving competency-based feedback were seen as clear, complete and useful. Most helpers expressed their gratitude for the feedback, which was perceived as nuanced, concrete and thus, helpful for their self-awareness. One helper reflected: "Very good, firstly, because I have reviewed everything I have learned in a short intensive time and then I hear the opinion of a person who is observing me. That's very useful for me." Other helpers commented, "I really needed this assessment," "I have no questions, everything was very clear," "Thank you for the experience."

At the start of the assessment, the rater observed tensions in some helpers due to unfamiliarity with the task and concerns about possible consequences. One helper noted, "Because there's a camera there and everything is being recorded, you might be a bit nervous." However, the rater reported that explaining the assessment's purpose appeared to reduce stress.

PM+ role-play instructions were perceived as understandable by the helpers. Helpers remarked, "It was pleasant. It was similar to the sessions we had previously held with the clients," "I was feeling relaxed, as I'm used to this." Regarding the ENACT role-play, some helpers expressed uncertainty about conducting open-ended conversations because they had been trained to follow the manualized PM+ intervention. For example, one helper asked, "Why should I do this now? Aren't we supposed to do something else first at this point"? Thus, active prompting by the actor was required to guide them through their assignment, occasionally reducing time available for them to respond and demonstrate their skills. Several helpers commented on this challenge: "In real sessions, I have more time. Here, everything went so fast, it was difficult to fit everything in," "With the fast pace, there were many things I had to consider in

order to carry out PM+ the way I learned," "He kept bringing up sensitive topics, which was not easy. I am glad that I was able to recognize this at the right moments and react accordingly."

According to the qualitative feedback from the rater, his general observations about helpers' competencies made during prior training and supervisions were largely confirmed by results of the EQUIP assessment, thus confirming accuracy of his prior judgments. EQUIP also helped correct inaccurate initial judgments, as some helpers exceeded or fell short of the rater's expectations in certain competencies.

Rater and actor saw great potential in the EQUIP package for competency monitoring and training adaptation as this standardized evaluation procedure effectively identified helpers' strengths and areas for improvement.

### Tool translation and cultural adaptation

Qualitative feedback from the rater and actor indicated that they and the helpers understood the Arabic EQUIP tool translations well. They noted that conducting competency assessments in the native language facilitated seamless role-playing, enhancing authenticity.

## Discussion

### Main findings

The main objective of this pilot study was to assess acceptability and preliminary utility of the Arabic version of the EQUIP tools in a pragmatic setting in Switzerland, within a PM+ trial. We demonstrated successful planning and implementation of the assessment using the EQUIP package. Regarding our second aim, results show that these tools can assess helpers' competencies and identify inter- and intra-individual strengths and areas for improvement. Arabic EQUIP tools demonstrate robustness, including for experienced and educated Arabic-speaking helpers. Thirdly, the assessments provide valuable information for conceptualization of training activities.

### Helpers' assessed performances

EQUIP assessments provided a differentiated overview of helpers' individual performances, highlighting their strengths and areas for improvement. Best results in the ENACT tool were obtained for items related to skills that are intensively exercised in PM+ training and supervision. Following items had scores that were two standard deviations above the average: Confidentiality, Psychoeducation, Non-Verbal Communication, Exploration & Normalization of Feelings and Verbal Communication.

However, some competencies were scored significantly low, two standard deviations below the average, such as Client's Explanation for Problem, Eliciting Feedback, and Coping & Prior Solutions. These competencies are not part of the training curriculum for helpers delivering PM+. This is in line with the finding that in regard to common therapeutic competencies, there are different approaches to their formulation, training and application across task-shifted interventions (Pedersen et al., 2020). Notably, trainers of the EQUIP Foundational Helping Skills Trainer's Curriculum (FHS) (Watts et al., 2021) identified competencies, including those mentioned above, as "more advanced" and "less relevant" for helpers (Pedersen et al., 2023b). Nonetheless, these competencies were included, both to collect comparable data for standardization of competency assessments and in line with recommendations from the EQUIP

developers. Their inclusion also allowed us to confirm expected competency gaps and to identify areas for future refinement of the training curriculum to achieve broader competency standards.

Assessment of Harm (item 4) was performed only moderately. While no more than seven out of 13 helpers inquired about risk or protective factors (basic skill), only one proposed developing a safety plan (advanced skill). Additionally, one helper lectured the client on religious grounds against self-harm, which was rated as unhelpful or potentially harmful behavior. These findings are surprising, given the strong emphasis placed on this competency during training and supervision. A systematic response plan for addressing suicide ideation, including specific referral procedures, was provided to helpers and regularly practiced prior to the assessment.

Similarly, another study confirms consistently low performances in this competency, likely due to its sensitive content (Pedersen et al., 2023b). Suicide globally constitutes a stigma associated taboo and is criminalized in many parts of the world (World Health Organization (WHO), 2021). Further, helpers are rarely confronted with clients with imminent thoughts of suicide, because people at high risk of suicide are usually screened out before being referred for the PM+ intervention. Thus, helpers may not be routinely habituated to such presentations. As this competency is critical for dealing with clients experiencing mental health problems, command of these skills should be ensured through regular training and reinforcement.

Collectively, these findings highlight the importance of competency-based training programs such as EQUIP-FHS (Watts et al., 2021; Pedersen et al., 2023b) as similarly demonstrated by Pedersen et al. (2023b), with EQUIP FHS ensuring helpers acquired proficiency and avoided harmful behaviors when applying common therapeutic skills. Furthermore, given the tertiary level education of the non-specialist helpers in this study also benefiting from these approaches (*i.e.*, assessments revealed a full range of high, low, and moderate skills on different competencies), EQUIP is likely just as suitable for well-educated versus less-educated trainee cohorts.

Moreover, future assessments might consider focusing on ENACT items that are most relevant to helpers, which could reduce assessment time. To note, the current version of the EQUIP PM+ tool (www.equipcompetency.org) is aligned to the ENACT tool used in this study, using a similar checkbox approach and assessment algorithm.

Despite overall strong performances on the PM+ specific assessment, the "Managing Problems" strategy emerged as a challenge for some helpers. The strategy's items accounted for the five lowest scoring items (items 2, 3, 4, 5, and 7), being recorded below the average, confirming previous findings that this strategy is particularly challenging (Van'T Hof et al., 2018; Gebrekristos et al., 2021).

This may be due to its comparatively high complexity as a seven-step strategy, requiring both close interaction and resistance to the helper's natural urge to give direct advice, while being capable of collaboratively identifying manageable problems, breaking them down into manageable elements, framing a problem statement, and exploring prospective solutions.

Helpers' better performance on the PM+ assessment than the ENACT tool could be due to differences in tool construction and instructions (*i.e.*, different rating procedures and presentation of attributes between the two assessments). Still, the helpers are trained to follow a detailed manual, but not for conducting a 15-minutes psychological interview as required in the ENACT role-play. Furthermore, PM+ competencies are more prominent in regular supervisions and would therefore be better recalled than ENACT competencies.

### Acceptance and qualitative evaluation of the rating protocol

Helpers, the rater and actor all accepted the rating protocol. Despite initial uncertainty, helpers followed the instructions and no helper withdrew, as observed in other studies (Pedersen et al., 2021). Helpers appreciated the individualized feedback, which as reported increased their self-awareness. However, it is important to investigate the impact of feedback on performance development through a longitudinal study design. Furthermore, challenges faced by the helpers when they had to speak in a relatively unstructured way during the ENACT role-play might be solved by further specifying instructions for the actor to efficiently assist in leading it.

EQUIP results aligned with the supervisor's perceptions, supporting its validity. Thus, EQUIP improves measurement in a standardized, communicable manner and appears to reduce judgment errors.

### Strengths and limitations

This is the first study examining an Arabic EQUIP tool version for PM+ in a HIC and the first study to use a PM+ tool covering all five sessions and its four core strategies. Another strength is the helpers' extensive experience with PM+ in a real-world setting and the investigators' profound theoretical and practical expertise with PM+ and EQUIP.

The post-treatment design only allowed for an interpretation of helpers' current competencies, and not of training or time effects. As the rater and actor in this study were also authors (and one was also involved in the helpers' training), objectivity might be limited, and confirmation and experimenter bias are possible. To control for these biases, assessments were videotaped for possible re-evaluation. However, exchanges between the rater and actor showed that ratings were consistent and therefore re-rating was not required. To note, videotaping could have increased stress and anxiety, negatively affecting performance of the helpers (Dickerson and Kemeny, 2004). On the other hand, it may also enhance performance as helpers strive to meet perceived expectations, known as the Hawthorne effect (McCambridge et al., 2014).

The assessment duration was 50 minutes, plus 5 minutes for preparation and 10 minutes for formulation and delivery of feedback. This effort may be unrealistic in resource-constrained real-world settings. To reduce time, implementers could limit the ENACT assessment to 10 minutes and focus on ENACT or PM+ items most relevant to their context.

Alternative assessment strategies could enhance EQUIP's feasibility in real-world settings. Use of audio recordings has proven to be feasible for ENACT ratings (Mwenge et al., 2022; Laurenzi et al., 2024) and has potential to reduce costs. Application of artificial intelligence and natural language processing show promise for automated assessment and feedback (Le Glaz et al., 2021; Malgaroli et al., 2023). Although these technologies are limited in capturing non-verbal cues, they could expand access to EQUIP. Further research is needed to assess their feasibility and ethical implications.

This study was conducted in a HIC and focused on the PM+ intervention, delivered in Arabic by experienced and educated helpers. This limits the generalizability of the findings to other settings, cultural contexts, or interventions, where helpers may have less formal education. The small sample size also restricted further statistical analyses.

### Future directions

Future research should consider longitudinal designs to examine training and time effects. Developing a predictive model incorporating helper characteristics might be likewise valuable. Further testing of EQUIP across diverse settings is also recommended, especially in real-world contexts where evidence-based psychological interventions are being implemented. Additionally, EQUIP assessments could serve as a certification test (Pedersen et al., 2023a, 2023b). This would require setting benchmarks and defining measures in case of failure (*e.g.*, refresher trainings). To understand the cost-effectiveness of EQUIP in the implementation of PM+, further research and replication are also required (Pedersen et al., 2023a).

Importantly, a recent study of helpers demonstrated that role-play assessments are better predictors of client outcomes than knowledge tests (Singla et al., 2023). However, the relationship between helpers' competencies and client health outcomes, treatment satisfaction, and quality of relationship warrants further investigation (Van Ginneken et al., 2013, 2021; Caulfield et al., 2019; Ottman et al., 2020).

### Conclusion

This study successfully assessed acceptability and preliminary utility of EQUIP in a pragmatic PM+ setting in Switzerland. EQUIP provides a nuanced and concise competency assessment in Arabic-speaking helpers, thus enhancing quality assurance and subsequent training measures. Outcomes suggest that EQUIP's competency assessment tools may be applied in larger studies.

**Open peer review.** To view the open peer review materials for this article, please visit http://doi.org/10.1017/gmh.2025.10023.

**Supplementary material.** The supplementary material for this article can be found at http://doi.org/10.1017/gmh.2025.10023.

**Data availability statement.** The data that support the findings of this study are available from the corresponding author, upon reasonable request.

**Acknowledgements.** This study was conducted within the STRENGTHS and EQUIP projects. We thank Karim Zagha and Hanan Zein Eddin for their support in translating the EQUIP tools, as well as Frezgi Gebrekristos for his collaboration on the development of the initial PM+ assessment tool, and Ibrahim Alhasan for performing final data verification. We are also grateful to Christian Gross, Antje Frey Nascimento, and Franz Neuberger for their valuable feedback on the initial draft of this manuscript.

**Author contribution.** The research was conceptualized by MH, AA, BAK, GAP, AS, RAB, and NM. Methodology development was a collaborative effort by MH, AA, BAK, GAP, AFA, CMS, AS, RAB, and NM. Formal analysis was performed by MH and AFA. Investigation and data collection were carried out by MH, AFA, and JS. The provision of resources necessary for the study was supported by MH, BAK, GAP, JS, and NM. Data curation and management were handled by MH and JS. The original draft of the manuscript was written by MH and AA and revised by GAP, RAB, and NM. All authors reviewed and commented on subsequent versions of the manuscript, and read and approved the final manuscript. Supervision was provided by AA, NM, RAB, and CMS. Project administration was managed by GAP, JS, NM, and MH. Finally, funding for the project was secured by BAK, AS, RAB, and NM.

**Financial support.** This study was supported by the Swiss State Secretariat for Education, Research, and Innovation under contract number 16.0205. The opinions expressed and arguments employed herein do not necessarily reflect

the official views of the Swiss Government. BAK, GAP, and AS were supported through USAID funding to the World Health Organization for the EQUIP initiative.

**Competing interests.** We have no conflicts of interest to disclose.

**Ethics statement.** This study was approved by the Ethics Committee of the Canton of Zurich (BASEC No. 2017-0117) and was conducted in accordance with the principles of the Declaration of Helsinki. Participation in the EQUIP competency assessments was a mandatory part of the participants' role as PM+ helpers delivering sessions to clients. All participants were fully informed of the aims and procedures of the study and provided written informed consent for the use of their data in research. Confidentiality and anonymity were strictly maintained, and all data were securely stored with access restricted to authorized personnel. The study complied with relevant Swiss regulations and institutional ethical standards.

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
