## [Reviewer Report]

This paper is really needed for PM+ practitioners, trainers/supervisors and implementors. I like the adaptation

made in PM+ assessments focusing on all 5 sessions and 4 strategies, and reducing 3 items to 1 item in Stress

management (focusing on slow breathing) from practical aspect. It looks like the foundational helping skills of the

helpers such as Verbal, non-verbal, confidentiality, psychoeducation and some PM+ skills has already been

improved before this study (validating the rater’s qualitative response). As being a PM+ trainer/supervisor/rater/

actor, I have some queries/ comments: 1. I suppose this study is done with individual PM+, not Group PM+.

Please mention it in the paper so that the reader can see it from individual perspective. 2. In the Method section,

while reading line 152, I imagine the helpers already have foundational skills and PM+ (as they have 1.5 years of

experience) prior to study, I am curious if the assessment (e.g ENACT) was done before the study? If so, what

were there pre-assessment? 2. In Measure section, line 158, who are the trained actor in this study (apart from

their involvement in adapting EQUIP tools). E.g their background? Line 160 (Can you add adapted case

vignettes in Supplementary) so that the readers like me know the background of cases. 3. Line 171, 172 and 173

- Please add one of the attributes as an example for Level 2, 3 and 4 like you have mentioned example for level

1, so that the readers know what these level looks like. 4. Line 202 - Clarify rating observed performance in this

line (direct rating? or rating the video recording?), although it is clear after reading line 236-248. 5. Line 293-296:

Would like to see examples/ attributes for each mentioned items. 6. Line 336-337: This needs to be mentioned in

the rater section (where there is little information about rater), as he has conducted trainings and supervisions as

well. 8. Challenges in Specific competencies: Line 389 - Can you give examples which attributes within Item 4?

This can help the readers (especially trainers) to know what to highlight during training. 9. Future directions: Line 467 - With regards to in real-world contexts, what do you all think about assessment using audio recording by the

rater from the real world setting so that it can be used for supervision (where direct supervision is costly)?

---

## [Reviewer Report]

OVERALL COMMENTS

• Well-structured and coherent rationale section

• Please adhere to the author guidelines for in-text referencing format

• Methods are very detailed; consider trimming some of the text to make the description more succinct as there are instances where there is repetition.

INTRODUCTION

• Kindly provide a reference for the PM+ intervention descriptions in Lines 51-54; Page 3

• Consider also providing contextual information (e.g. effect sizes) that strengthen the PM+ evidence base. Currently, the arguments for the interventions are purely descriptive in nature. It could also be worthwhile comparing its effectiveness with other task-shifted approaches.

• Reference the Kohrt et al (2020) paper in line 71 (Page 3) properly

• Be more specific on the names of the actual EQUIP tools that have been formally validated; the current description is vague

• The statement on cost analysis (lines 93-96; Page 4) seems out of sync. Kindly review this to ensure synergy with the preceding arguments.

METHODS

• Can you please explain the rationale for the helpers to have a higher education diploma as part of the inclusion criteria?

• Can you explain the rationale for the addition of two items to the PM+ tool

RESULTS

• Round age figures to one decimal.

• Given the higher educational attainment of the sample, I wonder how this affects the study’s comparability to other task-shifted interventions.

• Please summarize the descriptive results more.

• Perhaps you could focus on purely descriptive statistics given the small sample size, the use of bivariate tests does not seem appropriate.

• Consider providing some succinct verbatim quotes to support the qualitative outcomes.

DISCUSSION

• Excellent study findings from the onset of the discussion section.

• I am unsure if the comparison of the two tools is the right approach to interpreting the results; it seems slightly off-tangent to the study’s aims.

• Kindly refer to my comments regarding bivariate analysis

• The conclusion section is well done

---

## [Reviewer Report]

The manuscript is now much improved, kindly address the suggested comments. here are the specific comments:

Impact statement

Lines 49-50: The external applicability of the PM+ tools in low-income countries seems a bit of an overreach, considering the study used helpers with advanced education, which will not be applicable/transferable to low-income countries.

Introduction

• Page 3; lines 67-70, the statement, “In contrast to existing evidence-based brief psychological interventions, PM+ 68 combines the advantages of scalability, adaptability, and well-structured delivery.”. this statement is controversial and not entirely true as they are other brief, task-shifted interventions with excellent evidence of effectiveness and scalability. Kindly rephrase.

• Again, the edits implemented in Lines 71-76 are also not entirely accurate, although the authors are applauded for giving specific examples. Unfortunately, the statements are also hugely biased towards supporting their arguments, but do not necessarily capture the actual realities of task-shifting. For example, the Friendship Bench is available to all adults and has excellent evidence of clinical effectiveness and implementability.

• Well done for providing the empirical evidence of the effectiveness/efficacy of the PM+ intervention, a crucial element missing from the last draft.

• Lines 143-146: Consider moving the text on the cost analysis of the EQUIP-based approach, as it provides a strong rationale for the study.

Methods & results

• The training of highly educated helpers seems to counteract your arguments for the need of scalable and simple interventions in the introduction section.

• The tools and results are well described.

• The addition of verbatim quotes increased the qualitative results ' clarity.

Discussion & conclusion

• Although authors argue that some of the poorly rated competencies (Lines 514-517: Page 15) are not part of the helpers ' curriculum, more needs to be done to justify their inclusion in the first place. This is an area where a more nuanced discussion is required.

• The discussion on harm was concise and well contextualised; well done.

• The conclusion section is well-balanced

---

## [Editor Report]

Can you kindly address the minor comments? The manuscript is much improved from the last version and is almost there.